Rapid formation of secondary aerosol precursors from the autoxidation of C5-C8 n-1 aldehydes 2 Shawon Barua<sup>1,\*</sup>, Avinash Kumar<sup>1</sup>, Prasenjit Seal<sup>1</sup>, Siddharth Iyer<sup>1</sup>, and Matti Rissanen<sup>1,2,\*</sup> 3 <sup>1</sup>Aerosol Physics Laboratory, Physics Unit, Faculty of Engineering and Natural Sciences, 4 5 Tampere University, 33720 Tampere, Finland <sup>2</sup>Department of Chemistry, University of Helsinki, 00560 Helsinki, Finland 6 Correspondence: Shawon Barua (shawon.barua@tuni.fi) and Matti Rissanen 7 (matti.rissanen@tuni.fi) 8

## ABSTRACT

18

22

2526

27

28

Long chain aldehydes are common atmospheric constituents, and their gas-phase oxidation form low volatility condensable products leading to secondary organic aerosol. Although the oxidation of n-aldehydes initiated by OH radicals is dominated by aldehydic hydrogen abstraction, the non-aldehydic hydrogen abstractions tend to become competitive with the increase of aldehyde carbon chain length. Here, we experimentally investigated the oxidation of C<sub>5</sub>-C<sub>8</sub> n-aldehydes in variable reaction times (1-13 s) in a flow tube reactor coupled to a nitrate ion time-of-flight chemical ionization mass spectrometer (NO<sub>3</sub>-ToF-CIMS). Octanal produced highly oxygenated organic molecules (HOMs – low volatility products) with up to 7 O atoms within 1.0 s while the same level of oxygenation was acquired by pentanal within 2.3 s. In long reaction time (11–13 s) experiments, we observed HOMs with progressively more O atoms and higher intensity product signals with the increase of carbon atoms in the precursor aldehydes. Our experiments in the presence of high NO concentrations (2 ppb to 1 ppm) showed the formation of prominent highly oxygenated organonitrates along with the suppression of HOM accretion products. However, some enhancement in the monomeric HOMs even with 6 O atoms were seen under variable NO conditions. Results from hydrogen to deuterium (H/D) exchange experiments showed that the studied n-aldehydes undergo similar autoxidation mechanisms, but the reactivity and HOM formation potential increase with increasing carbon chain length.

#### INTRODUCTION

Secondary organic aerosol (SOA) refers to the aerosol material that is formed by the atmospheric gas-phase oxidation of volatile organic compounds (VOCs) (Seinfeld and Pandis, 2016; Ziemann and Atkinson, 2012; Kroll and Seinfeld, 2008). Atmospheric oxidation of

VOCs increases the oxygen to carbon ratios (O:C) in the oxidation products and can form highly oxygenated organic molecules (HOMs). These low volatility products are found to play a key role in the formation and growth of SOA (Ehn et al., 2014; Öström et al., 2017; Bianchi et al., 2019; Brean et al., 2019, 2020). The SOA is a dominant component of tropospheric fine particulate matter (Hallquist et al., 2009; Huang et al., 2014; Spracklen et al., 2011), influences oxidative capacity, local and global air quality, climate change, and human health (Hansen and Sato, 2001; Jacobson et al., 2000; Kanakidou et al., 2005; Zhang et al., 2014). Despite having significant attention due to its importance on SOA, its sources and formation processes are yet to be fully understood.

Aldehydes are common emissions in natural and polluted environments and have both biogenic and anthropogenic sources (Lipari et al., 1984; Schauer et al., 1999a, 1999b, 2001; Ciccioli et al., 1993; Carlier et al., 1986) and are also formed by chemical transformation of other VOCs, especially ozonolysis of alkenes (Calogirou et al., 1999). Atmospheric degradation of aldehydes is mainly governed by photolysis and the reaction with OH radicals in the daytime (Calvert et al., 2020; Mellouki et al., 2003, 2015). During nighttime, the reactions with NO<sub>3</sub> radicals are the dominant sink of aldehydes (Calvert et al., 2020). Although there are prior kinetic studies of reactions of *n*-aldehydes with OH radicals available in the literature (Aguirre et al., 2025; Castañeda et al., 2012; Iuga et al., 2010; S. Wang et al., 2015; Cassanelli et al., 2005; Albaladejo et al., 2002), they are limited to the initial steps of oxidation except recent studies (Barua et al., 2023; Yang et al., 2024) showing further oxidation steps leading to the formation of more functionalized products including HOMs.

It is well understood that the reaction of aldehydes with OH radicals is predominantly initiated by the abstraction of aldehydic hydrogen atom due to its weaker bond strength. However, with the increase of carbon chain length, the abstraction of other hydrogen atoms distant from the aldehydic moiety can also contribute to the overall oxidation process. In high  $NO_x$  (NO +  $NO_2$ ) condition, the aldehydic H abstraction mainly leads to the cleavage of that carbon (C1) by CO loss from acyl (RC(O)) intermediate or to the acyl peroxy (RC(O)OO) followed by  $CO_2$  loss from the subsequent acyloxy (RC(O)O) intermediate ultimately forming  $C_{n-1}$  aldehyde,  $C_{n-1}$  alkyl nitrate, and  $C_{n-1}$  alkoxy isomerization products (Chacon-Madrid et al., 2010; Rissanen et al., 2014; Vereecken & Peeters, 2009). Besides, it can also produce  $C_n$  peroxyacyl nitrates (PAN), a reservoir species for long-range transport of  $NO_x$  in the free troposphere, and  $C_n$  peroxy acids by the reaction of RC(O)OO intermediate with  $NO_2$  and  $HO_2$ , respectively (Calvert et al., 2020; Mellouki et al., 2003, 2015; Chacon-Madrid et al., 2010; Barua et al., 2023).

68 69

70

71 72

77

82

85

8990

Chacon-Madrid et al. (2010) have conducted a comparative study of the SOA yields in OH-initiated oxidation of n-aldehydes and n-alkanes under high NO $_x$  conditions. They reported near identical SOA yields from n-tridecanal and n-dodecane where the precursor alkane is having one less carbon than the precursor aldehyde. The finding was attributed to the formation of  $C_{12}$  alkoxy radical intermediate from both precursors undergoing similar subsequent reactions leading to SOA. This indicates that the dominant fate of aldehyde oxidation by OH leads to the fragmentation of its carbon backbone losing one carbon atom rather than producing functionalized products with the same number of carbon atoms as the parent molecule.

In low NO<sub>x</sub> conditions, previous experimental studies have shown that the abstraction of the aldehydic hydrogen can also lead to molecular functionalization forming HOMs (Rissanen et al., 2014; Ehn et al., 2014; Tröstl et al., 2016; Wang et al., 2021) and potentially promote the SOA yields. Recently, Barua et al. (2023) have studied hexanal OH oxidation reaction in detail using high level quantum chemical computations as well as experimental mass spectrometry technique. They showed that both the aldehydic and non-aldehydic H abstraction pathways can contribute to the functionalization of hexanal resulting in rapid formation of HOMs via autoxidation. The autoxidation process involves sequential unimolecular isomerization reactions of the peroxy radical (RO<sub>2</sub>) intermediates followed by repeated molecular O2 additions and increasing product O:C ratios (Rissanen et al., 2014, 2015; Crounse et al., 2013; Jokinen et al., 2014; Mentel et al., 2015; Berndt et al., 2015, 2016) without the intervention of any bimolecular reactions, e.g., RO<sub>2</sub> + RO<sub>2</sub>, RO<sub>2</sub> + NO<sub>x</sub>, RO<sub>2</sub> + HO<sub>x</sub> (OH and HO<sub>2</sub>), etc. Note that the reaction with excess O<sub>2</sub> molecules is a pseudo unimolecular process in the autoxidation sequence. Along the aldehydic H abstraction reaction route, the fastest isomerization (1,6 H-shift rate coefficient,  $k = 0.2 \text{ s}^{-1}$ ) of the RC(O)OO intermediate was shown to be competitive for autoxidation reaction chain propagation against any bimolecular reaction mediated RC(O)O fragmentation and thus keeps the carbon backbone of the precursor aldehyde intact (Barua et al., 2023). A non-aldehydic H abstraction from C4 was also seen to be competitive and its corresponding RO2 was shown to undergo a 1,6 H-shift reaction with the aldehydic H atom at a higher rate  $(k = 0.9 \text{ s}^{-1})$  than the H-shifts in RC(O)OO radical. Moreover, the kinetic modelling simulation on OH-initiated oxidation of hexanal conducted by Barua et al. (2023) showed that a detectable concentration (1.3 × 10<sup>4</sup> molec.cm<sup>-</sup> <sup>3</sup>) of O<sub>7</sub> HOMs were produced even in the presence of 1 ppb NO. Because the non-aldehydic H abstraction pathways are likely less prone to fragmentation and promote functionalization, the effect of carbon chain length of *n*-aldehydes on HOM yields is of great interest.

In this work, we experimentally studied the OH initiated autoxidation of C<sub>5</sub>–C<sub>8</sub> *n*-aldehydes in variable reaction times under atmospheric pressure and room temperature using state-of-the-art mass spectrometry technique. Besides, the reactions were studied in the presence of variable concentrations of NO to examine the effect of NO on the oxidation process. The study shows how the length of carbon chain in linear aldehydes directly affects the reactivity and functionalization of the molecules during their oxidation initiated by OH radicals.

#### METHODS

#### 2.1 Experimental setup

The gas-phase oxidation reactions of *n*-aldehydes with OH radicals were conducted in a flow reactor setup in the laboratory as shown in Fig. S1 in the Supplement. All the experiments were conducted at room temperature and 1 atm pressure of air. The precursor aldehydes were introduced to the reactor from their individual gas cylinders. The oxidant OH radicals were produced in situ by the reaction of tetramethylethylene (TME, C<sub>6</sub>H<sub>12</sub>) with ozone. We used an ozone generator that photolyzes zero-air by a mercury lamp (UVP, Analytik Jena) to provide ozone to the reactor while TME was supplied from a gas cylinder. The details of the chemicals and gas cylinders are given in Sect. S2 in the Supplement. The zero-air was produced by feeding compressed clean air to a zero-air generator (AADCO-737-15) which was also used as bath gas in the reactor maintaining a total sample flow of 8–10 slpm. The initial concentrations of reactant precursors were determined by controlling the individual gas flows using calibrated mass flow controllers (Alicat Scientific). An ozone analyzer (2B Technologies model 205) was used to measure the ozone concentrations.

We conducted the reactions at variable reaction times (short: 1–3 s, and long: 11–13 s). The experimental conditions are presented in Table 1. Long reaction time experiments were conducted using a borosilicate flow reactor (length: 100 cm, and i.d.: 4.7 cm) while a quartz flow reactor (length: 100 cm, and i.d.: 2.2 cm) was used for the short reaction time experiments. We utilized the full volume of the reactor to achieve a long reaction time. However, the short reaction times were achieved by controlling the distance between the mass spectrometer orifice and the position where the precursor aldehyde meets the oxidant OH inside the reactor. This was done by using a movable injector that brings the aldehyde of interest at variable positions inside the reactor. The shortest possible reaction time of an individual aldehyde was chosen by the detection of any HOMs from its oxidation initiated by OH radicals. In short reaction time

132

133

135

136137

138

139

140

141142

148

149150

151

experiments, the highest concentration of VOC (6.4 ppmv) was used for pentanal while the concentrations of other aldehydes remained within 1 ppmv. The VOC concentrations of 0.2-2.5 ppmv were used in long reaction time experiments. The other reactants including 43-97 ppbv of TME, and 77-295 ppbv of ozone were maintained nearly constant with respect to individual VOC in all experiment types (see Table 1). Among all the studied n-aldehyde systems, the lowest level of aldehyde and O<sub>3</sub> concentrations were maintained for heptanal oxidation experiment. This is because higher concentrations led to irrelevant products likely originating from the ozonolysis of heptanal stabilizers (see Fig. S2 in the Supplement). To observe the effect of NO<sub>x</sub> on the OH induced oxidation of *n*-aldehydes, we also conducted the experiments with variable NO concentrations (2–1000 ppbv). Additionally, one more set of experiments, hydrogen to deuterium (H/D) exchange, were conducted by the addition of D<sub>2</sub>O flow to the n-aldehyde OH oxidation reaction. These experiments give an estimate of the number of functional groups with labile H atoms (e.g., OH, OOH, and C(O)OOH) in the oxidation products. A near complete H/D conversion was confirmed by monitoring the reagent ion signals of HNO<sub>3</sub>NO<sub>3</sub><sup>-</sup> and (HNO<sub>3</sub>)<sub>2</sub>NO<sub>3</sub><sup>-</sup> fully converting to DNO<sub>3</sub>NO<sub>3</sub><sup>-</sup> and (DNO<sub>3</sub>)<sub>2</sub>NO<sub>3</sub><sup>-</sup> , respectively (see Figure S12 in the Supplement).

The oxidation products were detected using a nitrate ion time-of-flight chemical ionization mass spectrometer ( $NO_3^-$ -ToF-CIMS) as their  $NO_3^-$  adducts. A zero-air sheath flow of 20 slpm was provided to the chemical ionization inlet. The  $NO_3^-$  ions were produced from gas-phase  $HNO_3$  flow under soft X-ray exposure while being carried by  $N_2$  to the inlet. The mass spectrometric data were analyzed using the tofTools v6.03 package for MATLAB.

**Table 1**. The experimental conditions for OH initiated oxidation of studied C<sub>5</sub>–C<sub>8</sub> *n*-aldehydes.

| Experiment type                             | [VOC] | [TME] | [O <sub>3</sub> ] | [OH] <sup>≠</sup> | [NO]            | D <sub>2</sub> O | $\Delta t^{\ddagger}$ |
|---------------------------------------------|-------|-------|-------------------|-------------------|-----------------|------------------|-----------------------|
| VOC                                         | ppmv  | ppbv  | ppbv              | pptv              | ppbv            | $y/n^{\pm}$      | S                     |
| Short residence time                        |       |       |                   |                   |                 |                  |                       |
| Pentanal (C <sub>5</sub> H <sub>10</sub> O) | 6.4   | 48.2  | 295               | 4.4               | $N\!/A^\dagger$ | n                | 2.3                   |
| Hexanal (C <sub>6</sub> H <sub>12</sub> O)  | 1.0   | 43.2  | 225               | 3.4               | N/A             | n                | 1.1, 2.9              |
| Octanal (C <sub>8</sub> H <sub>16</sub> O)  | 0.72  | 48.2  | 208               | 3.1               | N/A             | n                | 1.0, 2.1              |
| Long residence time                         |       |       |                   |                   |                 |                  |                       |
| Pentanal (C <sub>5</sub> H <sub>10</sub> O) | 2.5   | 48.2  | 295               | 4.4               | N/A             | n                | 12.8                  |
| Hexanal (C <sub>6</sub> H <sub>12</sub> O)  | 1.0   | 43.2  | 225               | 3.4               | N/A             | n                | 11.5                  |
| Heptanal (C <sub>7</sub> H <sub>14</sub> O) | 0.15  | 96.5  | 77                | 1.2               | N/A             | n                | 12.8                  |
| Octanal (C <sub>8</sub> H <sub>16</sub> O)  | 0.72  | 48.2  | 208               | 3.1               | N/A             | n                | 12.8                  |
| <b>Experiments with NO</b>                  |       |       |                   |                   |                 |                  |                       |
| Pentanal (C <sub>5</sub> H <sub>10</sub> O) | 1.3   | 48.2  | 208               | 3.1               | 2-1000          | n                | 12.8                  |
| Hexanal (C <sub>6</sub> H <sub>12</sub> O)  | 1.0   | 43.2  | 225               | 3.4               | 2-200           | n                | 11.5                  |
| Octanal (C <sub>8</sub> H <sub>16</sub> O)  | 0.72  | 48.2  | 208               | 3.1               | 2-1000          | n                | 12.8                  |
| Experiments with D <sub>2</sub> O           |       |       |                   |                   |                 |                  |                       |
| Pentanal (C <sub>5</sub> H <sub>10</sub> O) | 2.5   | 48.2  | 295               | 4.4               | N/A             | y                | 12.8                  |
| Hexanal (C <sub>6</sub> H <sub>12</sub> O)  | 1.0   | 43.2  | 225               | 3.4               | N/A             | у                | 11.5                  |
| Heptanal (C <sub>7</sub> H <sub>14</sub> O) | 0.10  | 96.5  | 77                | 1.2               | N/A             | y                | 12.8                  |
| Octanal (C <sub>8</sub> H <sub>16</sub> O)  | 0.72  | 48.2  | 208               | 3.1               | N/A             | y                | 12.8                  |

<sup>153</sup> 

The initial OH concentrations were calculated using bimolecular rate coefficients  $k_{O_3-TME}=1.5\times10^{-15}~{\rm cm^3~s^{-1}}, k_{OH-TME}=1.0\times10^{-10}~{\rm cm^3~s^{-1}}$  (Manion et al., 2015), and the expression  $[OH]=(k_{O_3-TME}*[O_3])/k_{OH-TME}$ . 154

<sup>155</sup> 

<sup>&</sup>lt;sup>†</sup> Not applicable (N/A). <sup>±</sup> D<sub>2</sub>O added = y, not added = n. <sup>‡</sup> Reaction time ( $\Delta t$ ). 156

159

167

173174

177178

180

182

#### RESULTS AND DISCUSSION

## 3.1 Detection of HOM in short to long reaction time experiments

In this section, we discuss how early HOMs formed and how they evolved with the progress of reaction time in different n-aldehyde OH oxidation experiments. The hexanal OH oxidation spectra included in Figures 1–2 are reproduced from our previous study (Barua et al., 2023) for comparison with the other aldehydes. Figure 1 shows the results from short reaction time ( $\Delta t$  = 1-3 s) experiments. We observed the formation of O<sub>6</sub> and O<sub>7</sub> HOMs within 1.0, 1.1, and 2.3 s reaction times in the oxidation experiments of octanal, hexanal, and pentanal, respectively. It is essential to mention that with the increase of number of carbon atoms in the studied aldehydes, the required precursor concentrations for first HOM appearance decreased (from 6.4 ppm pentanal, 1 ppm hexanal to 0.72 ppm octanal). This shows a clear effect of carbon chain length on the reactivity of linear aldehydes towards HOM formation. Because the oxidation process of n-aldehydes ( $C_nH_{2n}O$ ) with OH is initiated by the abstraction of a H atom (aldehydic or non-aldehydic), the first formed acyl (or alkyl) peroxy radical  $C_nH_{2n-1}O_3$  contains an odd number of oxygen atoms. If autoxidation outcompetes any other bimolecular reactions (e.g., RO<sub>2</sub> + RO<sub>2</sub>, RO<sub>2</sub> + HO<sub>2</sub>, etc.), the product spectrum will be mostly dominated by odd number of oxygen containing products. In all studied n-aldehyde systems, the intensity of O<sub>6</sub> HOM is higher than that of O<sub>7</sub> HOM (Figures 1–2). The formation of C<sub>n</sub>H<sub>2n-1</sub>O<sub>6</sub> peroxy radical indicates that the process certainly involves a bimolecular reaction step. In hexanal OH oxidation, Barua et al. (2023) computationally showed that the formation of  $C_6H_{11}O_5$  peroxy radical via autoxidation is very fast while the subsequent isomerization reaction leading to  $C_6H_{11}O_7$  is slower. The same is likely to hold true for other aldehydes and it is expected that the C<sub>n</sub>H<sub>2n-1</sub>O<sub>5</sub> peroxy radical undergoes a bimolecular reaction converting it to C<sub>n</sub>H<sub>2n-1</sub>O<sub>4</sub> alkoxy radical, followed by a H-shift and subsequent O2 addition reactions producing the dominant C<sub>n</sub>H<sub>2n-1</sub>O<sub>6</sub> HOM (see Fig. S5–S9 in the Supplement). As the reaction time increased, we observed the formation of monomeric HOM up to O<sub>7</sub> and accretion products up to O<sub>10</sub> composition within 2.9 s in hexanal oxidation while in the case of octanal, monomeric O<sub>8</sub> HOM and accretion products up to O<sub>10</sub> appeared within 2.1 s reaction time (see Figure 1 D–E).

**Figure 1.** Nitrate chemical ionization mass spectra of OH initiated oxidation of n-aldehydes showing the formation of HOMs in different reaction times ( $\Delta t$ ): 2.3 s – pentanal in black (A), 1.1 and 2.9 s – hexanal in blue (B and D), and 1.0 and 2.1 s – octanal in purple (C and E). The product peaks are labelled with the exclusion of NO<sub>3</sub><sup>-</sup> ion attachment in their compositions. The accretion product region is highlighted in light gold background. The accretion products labeled with nominal mass/charge in dark red ( $C_{n+3}H_{2n+4}O_{7-8}$ ) are related to the TME-derived peroxy radical  $C_3H_5O_3$ .

In long reaction time (11–13 s) experiments, we observed higher intensities of product signals (see Figure 2) in comparison to their intensities in the short reaction time experiments - as expected. For pentanal, Figure 2A shows that HOM accretion products up to  $O_{10}$  formed

https://doi.org/10.5194/egusphere-2025-5207 Preprint. Discussion started: 24 October 2025 © Author(s) 2025. CC BY 4.0 License.

within 12.8 s reaction time which were not seen in the short reaction time experiment (2.3 s). A lower precursor concentration, 2.5 ppm of pentanal in long reaction time experiment compared to earlier 6.4 ppm in short reaction time, was sufficient to produce the observed HOMs in this case. A close observation of  $C_6$ – $C_8$  n-aldehyde oxidation spectra (Figure 2B–D) reveals that HOM accretion products up to  $O_{11}$  formed withing 11–13 s reaction time under the experimental conditions. In all n-aldehyde experiments, we also observed the accretion products resulting from different combinations of aldehyde-derived peroxy radicals  $C_nH_{2n-1}O_{6-8}$  and TME-derived peroxy radical  $C_3H_5O_3$  (see Fig. S11 in the Supplement for details) which are marked with dark red arrows. Figure 2D implies that the highest oxygenation ( $C_8H_{15}O_9$ ) in the monomeric HOM products is associated with octanal while heptanal produced most oxygenated products are  $C_7H_{12-14}O_8$  (see Figure 2C). In the case of pentanal and hexanal, monomeric HOMs are limited to seven oxygen atoms (see Figures 2A–B). All in all, we notice a near identical distribution of oxidation products in the experiments with all  $C_5$ – $C_8$  n-aldehydes. However, the tendency of oxidation gets faster and advances to higher oxygenated products when the carbon chain length increases.

**Figure 2.** Nitrate chemical ionization mass spectra of OH initiated oxidation of n-aldehydes in 11-13 s residence times: pentanal in black (A), hexanal in blue (B), heptanal in red (C), and octanal in purple (D). The product peaks are labelled with the exclusion of  $NO_3^-$  ion attachment in their compositions. The accretion product region is highlighted in light gold background. The accretion products marked by dark red arrows ( $C_{n+3}H_{2n+4}O_{7-9}$ ) are related to the TME-derived peroxy radical  $C_3H_5O_3$ .

The bar plot (see Figure 3) compares the normalized signal intensities of major oxidation products from different n-aldehydes. It clearly shows that the yield of higher oxygenated products increases as we move from pentanal to octanal. In the accretion product regime, the intensities of  $O_9$ – $O_{10}$  products in octanal are lower than that of hexanal which is

also reflected in their dimer to monomer ratios with octanal being  $8.8 \times 10^{-2}$  and hexanal being  $2.5 \times 10^{-1}$ . The lower ratio for octanal compared to hexanal can be caused by the somewhat lower oxidant loading in octanal oxidation experiment (3.1 versus 3.4 pptv of initial OH in octanal and hexanal, respectively). This can also lie in the variation of  $RO_2 + RO_2$  reaction rate coefficients forming the accretion products  $(RO_2 + RO_2 \rightarrow ROOR + O_2)$  which is highly dependent on specific  $RO_2$  structures (Berndt et al., 2018; Shallcross et al., 2005).

**Figure 3.** Distribution of major oxidation products ( $O_5$ – $O_8$  in monomeric regime with white background, and  $O_9$ – $O_{10}$  in accretion product regime with light gold background) in pentanal, hexanal, and octanal oxidation initiated by OH radical. In y-axis, the numbers are the cumulative sum of normalized intensities of products with the same oxygen number. Reaction time,  $\Delta t = 11-13$  s.

# 3.2 Experiments in the presence of NO

It has been widely acknowledged that the formation HOM is suppressed in high  $NO_x$  conditions (Praske et al., 2018; Wildt et al., 2014; McFiggans et al., 2019; Pullinen et al., 2020) and so as the SOA yields, especially by halting the formation of HOM accretion products which are known to be extremely condensable (Pullinen et al., 2020; Kirkby et al., 2016). However, other studies have shown that NO can also enhance HOM formation by producing reactive RO radicals that can propagate autoxidation (Barua et al., 2025; Kang et al., 2025; Rissanen, 2018; Z. Wang et al., 2021; Nie et al., 2023). In our different n-aldehyde oxidation experiments in the presence of variable concentrations of NO, we observed the general tendency of dropping of the accretion product signals ( $C_{2n}H_{4n-2}O_z$ ) as expected (see Figures 4 and 5). Figure 4 represents the mass spectra recorded in the experiments without (in blue) and with the presence of 100 ppb NO (in red). At this condition, most of the monomeric products are quenched at different extents alongside the formation of organonitrates while the accretion products are

quenched nearly completely. Interestingly, we observed some enhancement in the intensities of  $O_4$  products in the case of hexanal and octanal (Figure 4 B–C) under 100 ppb NO. A closer look at Figure 5A reveals that the intensities of closed-shell product signals  $C_5H_8O_{4-6}$  somewhat increased at 2 ppb NO condition which then started to decrease with the injection of higher NO of 50 ppb and above in pentanal oxidation. In both pentanal and hexanal oxidations, the dominant  $O_6$  peroxy radicals ( $C_5H_9O_6$  and  $C_6H_{11}O_6$  respectively, blue markers) seem to retain their initial intensities under up to 50 ppb of NO (Figure 5 A–B).

**Figure 4.** Overlaid nitrate chemical ionization mass spectra of OH initiated oxidation of n-aldehydes without (in blue) and with the presence of 100 ppb of NO (in red): pentanal (A), hexanal (B), and octanal (C). The product peaks are labelled with the exclusion of  $NO_3^-$  ion attachment in their compositions. The accretion product region is highlighted in light gold background. The accretion products marked by dark red arrows ( $C_{n+3}H_{2n+4}O_{7-9}$ ) are related to the TME-derived peroxy radical  $C_3H_5O_3$ . The organonitrates (in the presence of NO) are labelled with the extension NO and marked with purple arrows. Reaction time,  $\Delta t = 11-13$  s.

On the other hand, in octanal oxidation, the dominant  $O_6$  peroxy radical ( $C_8H_{15}O_6$ ) seems to maintain almost its initial level of signal intensity even under 100 ppb of NO (see Figure 5C). These observations indicate that the suppressing effect of NO on the yields of HOMs in n-aldehyde oxidation is perturbed with the increase of carbon chain length of the precursor aldehyde. Also, in all the studied n-aldehydes, although highly oxygenated products are suppressed under higher NO concentrations, we see some enhancement in the early oxygenated closed shell products ( $O_4$ – $O_5$ , and even  $O_6$  product in pentanal) under relatively lower NO concentrations. It should be noted that the formation of organonitrates with chemical composition  $C_nH_{2n-1}O_zNO$  strongly supports our assignment of reactive peroxy radical intermediates  $C_nH_{2n-1}O_z$ .

**Figure 5.** The response (in normalized counts  $s^{-1}$ ) of different oxidation product signals including monomeric HOMs, organonitrates (green markers), and HOM accretion products (black markers) as a function of NO concentrations in OH initiated oxidation of n-aldehydes: pentanal (A), hexanal (B), and octanal (C). Note the logarithmic scale (x-axis) in panels A and C. The orange rectangles include the enhanced non-nitrogen containing product signals: O<sub>4</sub>–O<sub>6</sub> closed-shell products from pentanal at 2 ppb NO (A), O<sub>4</sub> closed-shell products from hexanal and octanal at 100 ppb NO (B and C). Reaction time,  $\Delta t = 11-13$  s.

## 3.3 D<sub>2</sub>O experiments

With the addition of  $D_2O$  in the OH initiated oxidation experiments of n-aldehydes, we observed a shift in individual product signals in the mass spectra equivalent to the number of exchangeable H atoms in the product structures (see Figure 6). This provides additional insight into the product identities in terms of the total number of OH, OOH, and (or) C(O)OOH groups present in their molecular structures. Figure 6 shows that the oxidation products with same number of oxygen atoms in all the studied n-aldehydes undergo identical mass shifts (H/D exchange) in the presence of  $D_2O$ . This observation indicates that the autoxidation mechanism derived by Barua et al. (2023) for hexanal OH oxidation (see Fig. S3 in the Supplement) is directly applicable to the other linear aldehydes studied here and thus produces similar product structures.

**Figure 6.** Overlaid nitrate chemical ionization mass spectra of OH initiated oxidation of n-292 aldehydes without (in blue) and with the presence of D<sub>2</sub>O (in black): pentanal (A), hexanal (B), 293 heptanal (C), and octanal (D). In panel (A), the label (DNO<sub>3</sub>)<sub>2</sub>NO<sub>3</sub><sup>-</sup> is assigned to the deuterated 294 295 nitric acid trimer reagent signal. The product peaks are labelled with the exclusion of NO<sub>3</sub><sup>-</sup> ion 296 attachment in their compositions and the numbers on the blue arrows indicate the individual counts of mass shift during H/D exchange. The accretion product region is highlighted in light 297 gold background. 298 The original mechanism is extended to HOMs up to nine oxygen atoms and presented in Fig. 299 S4 in the Supplement. The likely formation process of HOM accretion products (C<sub>2n</sub>H<sub>4n-2</sub>O<sub>9-</sub> 300 11) is shown in Fig. S10 in the Supplement. The proposed structures of O<sub>5</sub> closed-shell products, 301 O<sub>6</sub>-O<sub>8</sub> monomeric HOMs as well as O<sub>9-11</sub> HOM accretion products (see Fig. S4-S10 in the 302 303 Supplement) agree with the H/D exchange experiments in terms of their 2-4 units of mass 304 shifts in the respective *n*-aldehyde mass spectra (see Fig. 6).

## 3.4 Atmospheric implications

307

308309

315

319320

322

Ambient concentration of individual longer chain aldehyde ( $\geq C_5$ ) can vary from sub-ppb to several ppb depending on time and location (Ma et al., 2019; Li et al., 2018; Duan et al., 2008; Williams et al., 1996). A total concentration of C<sub>6</sub>–C<sub>10</sub> n-aldehydes in Monti Cimini Forest in Italy was measured to be 8.8 ppb (Ciccioli et al., 1993). In indoor air, the concentration can be significantly higher, even around 50 ppb (Birmili et al., 2022). Atmospheric lifetime of naldehydes due to their reactivity with OH radicals is generally less than 10 h (Aguirre et al., 2025; Albaladejo et al., 2002). Because of their significant photochemical ozone formation potential (Aguirre et al., 2025; Jenkin et al., 2017), they are good candidates for generating photochemical smog in NO<sub>x</sub> rich polluted urban atmosphere. On the other hand, previous studies have shown that atmospheric oxidation products of longer chain n-aldehydes are direct contributors to the formation of SOA (Chacon-Madrid et al., 2010; Fan et al., 2024). Here, we demonstrated that the studied C<sub>5</sub>–C<sub>8</sub> n-aldehydes can rapidly form HOM via autoxidation initiated by OH radicals, and the length of carbon chain controls the efficiency of the process. Therefore, with the increase of carbon chain length in *n*-aldehydes, the fast formation of HOMs is expected to take part in the early stages of gas-to-particle formation and growth contributing to atmospheric SOA. Our experiments in the presence of NO showed a general decreasing trend of HOM accretion products with increasing NO but also formed the corresponding highly oxygenated organic nitrates (HOM-ONs). In our study with pentanal at 2 ppb NO condition, some enhancement with oxidation products up to six O atoms was also seen. Pullinen et al.

332

333

334335

336

337

338339

340

341

342343

344

345

346347

348 349

350 351

352 353

354

355356

(2020) showed that both HOMs and HOM-ONs originated from the same peroxy radicals with
more than six O atoms condensed on particles by about 50% and those with more than eight O
atoms condensed by about 100% to form SOA in monoterpene photooxidation. Moreover,
other reports show that HOM-ONs originated from different VOCs can contribute to low
volatility products (Barua et al., 2025) and thereby particle growth and aerosol mass loading
(W. Huang et al., 2019; Lee et al., 2016; Fry et al., 2014).

## CONCLUSIONS

This study represents the significance of longer chain linear aldehydes ( $\geq C_5$ ), key components of atmospherically abundant oxygenated volatile organic compounds (OVOCs), in rapid formation of HOMs upon atmospheric oxidation and their potential contribution to atmospheric SOA. Among the studied C<sub>5</sub>–C<sub>8</sub> n-aldehydes, the fastest HOM formation is associated with octanal forming O<sub>6</sub> and O<sub>7</sub> HOMs within 1.0 s reaction time with low precursor loading. Pentanal and hexanal formed HOMs with the same number of oxygen atoms as early as 2.3 and 1.1 s, respectively but with higher precursor loadings (i.e., 6.4 ppm pentanal and 1.0 ppm hexanal compared to 0.72 ppm octanal). The highest oxygenated monomeric HOM with 9 oxygen atoms was formed from octanal whereas the numbers are up to 8 oxygen atoms for heptanal and 7 oxygen atoms for both pentanal and hexanal oxidation initiated by OH radicals within 13 s reaction time. Although the highest precursor concentration (6.4 ppm) was required for the first appearance of detectable HOM in short reaction time experiment with pentanal, a lower concentration (2.5 ppm) was used to obtain its observed mass spectrum in the long reaction experiment. The HOM accretion products with up to 11 oxygen atoms were observed in C<sub>6</sub>–C<sub>8</sub> n-aldehyde oxidation experiments while they were limited to maximum 10 oxygen atoms in pentanal case. We also observed the trend of increased oxidation products signal intensities with the increase of carbon chain length. In all studied systems, the dominant product signals are O<sub>6</sub> HOMs with the O<sub>5</sub> HOMs being the second dominant ones. Previous mechanistic understanding (Barua et al., 2023) as well as current experimental observations reveal that autoxidation process forming O<sub>5</sub> RO<sub>2</sub> in n-aldehydes is very fast while the subsequent unimolecular rearrangements of the RO2 intermediates are in competition with bimolecular reactions. The experiments in the presence of high NO concentrations (50 ppb and above) formed highest intensity HOM-ONs with the expense of HOM accretion products. However, some enhancements with low oxygenated closed-shell products were also seen under various NO conditions. The results of hydrogen to deuterium (H/D) exchange experiments with

- identical mass shifts in the oxidation products of all studied n-aldehydes imply that the
- autoxidation mechanism established for hexanal (Barua et al., 2023) is valid for other n-
- aldehydes. Therefore, accounting for linear aldehydes and their atmospheric oxidation with
- increasing importance to longer carbon chain length as a direct source of condensable materials
- even under moderately polluted urban areas is essential.

#### 362 • SUPPLEMENT

- Details about the experimental setup; chemicals and gas cylinders; mechanistic details of the
- oxidation steps; and mass spectra of H/D exchange experiments.

## **365** • AUTHOR CONTRIBUTIONS

- Conceptualization: MR, SB, AK; data curation: SB, AK; formal analysis: SB; investigation:
- SB, AK, PS, SI, MR; methodology: SB, AK; writing (original draft preparation): SB; writing
- (review and editing): SB, AK, PS, SI, MR; funding acquisition: SI, MR.

## **COMPETING INTERESTS**

The authors declare that they have no conflict of interest.

#### ACKNOWLEDGEMENTS

- This project has received funding from the European Research Council under the European
- Union's Horizon 2020 research and innovation programme under Grant No. 101002728 (ERC
- Consolidator Grant Project ADAPT) and from the European Union's horizon Europe research
- and innovation programme under Grant No. 101096133 (PAREMPI: particle emission
- prevention and impact: from real-world emissions of traffic to secondary PM of 585 urban air).
- The support from the Research Council of Finland (331207, 336531, 346373, 347775, 353836,
- and 355966) and Doctoral school of the Faculty of Engineering and Natural Sciences of
- Tampere University are greatly appreciated. We thank the tofTools team for providing the data
- analysis program.

371

## REFERENCES

- Aguirre, F., Lugo G, P. L., Straccia C, V. G., Teruel, M. A., & Blanco, M. B. (2025). Atmospheric oxidation of
- long chain aldehydes: OH and Cl reactivity, mechanisms and environmental impact. Atmospheric
- Environment, 360, 121429. https://doi.org/10.1016/j.atmosenv.2025.121429

| 385 | Albaladejo, J., Ballesteros, B., Jiménez, E., Martín, P., & Martínez, E. (2002). A PLP-LIF kinetic study of the                                                                                                                                                                                                                                                                                                                                                                                                                                                                                                                                                                                                                                                                                                                                                                                                                                                                                                                                                                                                                                                                                                                                                                                                                                                                                                                                                                                                                                                                                                                                                                                                                                                                                                                                                                                                                                                                                                                                                                                                                  |
|-----|----------------------------------------------------------------------------------------------------------------------------------------------------------------------------------------------------------------------------------------------------------------------------------------------------------------------------------------------------------------------------------------------------------------------------------------------------------------------------------------------------------------------------------------------------------------------------------------------------------------------------------------------------------------------------------------------------------------------------------------------------------------------------------------------------------------------------------------------------------------------------------------------------------------------------------------------------------------------------------------------------------------------------------------------------------------------------------------------------------------------------------------------------------------------------------------------------------------------------------------------------------------------------------------------------------------------------------------------------------------------------------------------------------------------------------------------------------------------------------------------------------------------------------------------------------------------------------------------------------------------------------------------------------------------------------------------------------------------------------------------------------------------------------------------------------------------------------------------------------------------------------------------------------------------------------------------------------------------------------------------------------------------------------------------------------------------------------------------------------------------------------|
| 386 | atmospheric reactivity of a series of C4-C7 saturated and unsaturated aliphatic aldehydes with OH.                                                                                                                                                                                                                                                                                                                                                                                                                                                                                                                                                                                                                                                                                                                                                                                                                                                                                                                                                                                                                                                                                                                                                                                                                                                                                                                                                                                                                                                                                                                                                                                                                                                                                                                                                                                                                                                                                                                                                                                                                               |
| 387 | Atmospheric Environment, 36(20), 3231–3239. https://doi.org/10.1016/S1352-2310(02)00323-0                                                                                                                                                                                                                                                                                                                                                                                                                                                                                                                                                                                                                                                                                                                                                                                                                                                                                                                                                                                                                                                                                                                                                                                                                                                                                                                                                                                                                                                                                                                                                                                                                                                                                                                                                                                                                                                                                                                                                                                                                                        |
| 388 | Barua, S., Iyer, S., Kumar, A., Seal, P., & Rissanen, M. (2023). An aldehyde as a rapid source of secondary                                                                                                                                                                                                                                                                                                                                                                                                                                                                                                                                                                                                                                                                                                                                                                                                                                                                                                                                                                                                                                                                                                                                                                                                                                                                                                                                                                                                                                                                                                                                                                                                                                                                                                                                                                                                                                                                                                                                                                                                                      |
| 389 | aerosol precursors: Theoretical and experimental study of hexanal autoxidation. Atmospheric                                                                                                                                                                                                                                                                                                                                                                                                                                                                                                                                                                                                                                                                                                                                                                                                                                                                                                                                                                                                                                                                                                                                                                                                                                                                                                                                                                                                                                                                                                                                                                                                                                                                                                                                                                                                                                                                                                                                                                                                                                      |
| 390 | Chemistry and Physics, 23(18), 10517-10532. https://doi.org/10.5194/acp-23-10517-2023                                                                                                                                                                                                                                                                                                                                                                                                                                                                                                                                                                                                                                                                                                                                                                                                                                                                                                                                                                                                                                                                                                                                                                                                                                                                                                                                                                                                                                                                                                                                                                                                                                                                                                                                                                                                                                                                                                                                                                                                                                            |
| 391 | Barua, S., Kumar, A., Seal, P., Bezaatpour, M., Jha, S., Myllys, N., Iyer, S., & Rissanen, M. (2025). Rapid                                                                                                                                                                                                                                                                                                                                                                                                                                                                                                                                                                                                                                                                                                                                                                                                                                                                                                                                                                                                                                                                                                                                                                                                                                                                                                                                                                                                                                                                                                                                                                                                                                                                                                                                                                                                                                                                                                                                                                                                                      |
| 392 | formation of aerosol precursors from the autoxidation of aromatic carbonyls and the remarkable                                                                                                                                                                                                                                                                                                                                                                                                                                                                                                                                                                                                                                                                                                                                                                                                                                                                                                                                                                                                                                                                                                                                                                                                                                                                                                                                                                                                                                                                                                                                                                                                                                                                                                                                                                                                                                                                                                                                                                                                                                   |
| 393 | enhancing influence of NO addition. In Review. https://doi.org/10.21203/rs.3.rs-7332278/v1                                                                                                                                                                                                                                                                                                                                                                                                                                                                                                                                                                                                                                                                                                                                                                                                                                                                                                                                                                                                                                                                                                                                                                                                                                                                                                                                                                                                                                                                                                                                                                                                                                                                                                                                                                                                                                                                                                                                                                                                                                       |
| 394 | Berndt, T., Richters, S., Jokinen, T., Hyttinen, N., Kurtén, T., Otkjær, R. V., Kjaergaard, H. G., Stratmann, F.,                                                                                                                                                                                                                                                                                                                                                                                                                                                                                                                                                                                                                                                                                                                                                                                                                                                                                                                                                                                                                                                                                                                                                                                                                                                                                                                                                                                                                                                                                                                                                                                                                                                                                                                                                                                                                                                                                                                                                                                                                |
| 395 | Herrmann, H., Sipilä, M., Kulmala, M., & Ehn, M. (2016). Hydroxyl radical-induced formation of                                                                                                                                                                                                                                                                                                                                                                                                                                                                                                                                                                                                                                                                                                                                                                                                                                                                                                                                                                                                                                                                                                                                                                                                                                                                                                                                                                                                                                                                                                                                                                                                                                                                                                                                                                                                                                                                                                                                                                                                                                   |
| 396 | highly oxidized organic compounds. Nature Communications, 7(1), 13677.                                                                                                                                                                                                                                                                                                                                                                                                                                                                                                                                                                                                                                                                                                                                                                                                                                                                                                                                                                                                                                                                                                                                                                                                                                                                                                                                                                                                                                                                                                                                                                                                                                                                                                                                                                                                                                                                                                                                                                                                                                                           |
| 397 | https://doi.org/10.1038/ncomms13677                                                                                                                                                                                                                                                                                                                                                                                                                                                                                                                                                                                                                                                                                                                                                                                                                                                                                                                                                                                                                                                                                                                                                                                                                                                                                                                                                                                                                                                                                                                                                                                                                                                                                                                                                                                                                                                                                                                                                                                                                                                                                              |
| 398 | Berndt, T., Richters, S., Kaethner, R., Voigtländer, J., Stratmann, F., Sipilä, M., Kulmala, M., & Herrmann, H.                                                                                                                                                                                                                                                                                                                                                                                                                                                                                                                                                                                                                                                                                                                                                                                                                                                                                                                                                                                                                                                                                                                                                                                                                                                                                                                                                                                                                                                                                                                                                                                                                                                                                                                                                                                                                                                                                                                                                                                                                  |
| 399 | (2015). Gas-Phase Ozonolysis of Cycloalkenes: Formation of Highly Oxidized RO <sub>2</sub> Radicals and Their                                                                                                                                                                                                                                                                                                                                                                                                                                                                                                                                                                                                                                                                                                                                                                                                                                                                                                                                                                                                                                                                                                                                                                                                                                                                                                                                                                                                                                                                                                                                                                                                                                                                                                                                                                                                                                                                                                                                                                                                                    |
| 400 | Reactions with NO, NO $_2$ , SO $_2$ , and Other RO $_2$ Radicals. The Journal of Physical Chemistry A,                                                                                                                                                                                                                                                                                                                                                                                                                                                                                                                                                                                                                                                                                                                                                                                                                                                                                                                                                                                                                                                                                                                                                                                                                                                                                                                                                                                                                                                                                                                                                                                                                                                                                                                                                                                                                                                                                                                                                                                                                          |
| 401 | 119(41), 10336–10348. https://doi.org/10.1021/acs.jpca.5b07295                                                                                                                                                                                                                                                                                                                                                                                                                                                                                                                                                                                                                                                                                                                                                                                                                                                                                                                                                                                                                                                                                                                                                                                                                                                                                                                                                                                                                                                                                                                                                                                                                                                                                                                                                                                                                                                                                                                                                                                                                                                                   |
| 402 | Berndt, T., Scholz, W., Mentler, B., Fischer, L., Herrmann, H., Kulmala, M., & Hansel, A. (2018). Accretion                                                                                                                                                                                                                                                                                                                                                                                                                                                                                                                                                                                                                                                                                                                                                                                                                                                                                                                                                                                                                                                                                                                                                                                                                                                                                                                                                                                                                                                                                                                                                                                                                                                                                                                                                                                                                                                                                                                                                                                                                      |
| 403 | Product Formation from Self- and Cross-Reactions of RO <sub>2</sub> Radicals in the Atmosphere. Angewandte                                                                                                                                                                                                                                                                                                                                                                                                                                                                                                                                                                                                                                                                                                                                                                                                                                                                                                                                                                                                                                                                                                                                                                                                                                                                                                                                                                                                                                                                                                                                                                                                                                                                                                                                                                                                                                                                                                                                                                                                                       |
| 404 | Chemie International Edition, 57(14), 3820–3824. https://doi.org/10.1002/anie.201710989                                                                                                                                                                                                                                                                                                                                                                                                                                                                                                                                                                                                                                                                                                                                                                                                                                                                                                                                                                                                                                                                                                                                                                                                                                                                                                                                                                                                                                                                                                                                                                                                                                                                                                                                                                                                                                                                                                                                                                                                                                          |
| 405 | Bianchi, F., Kurtén, T., Riva, M., Mohr, C., Rissanen, M. P., Roldin, P., Berndt, T., Crounse, J. D., Wennberg,                                                                                                                                                                                                                                                                                                                                                                                                                                                                                                                                                                                                                                                                                                                                                                                                                                                                                                                                                                                                                                                                                                                                                                                                                                                                                                                                                                                                                                                                                                                                                                                                                                                                                                                                                                                                                                                                                                                                                                                                                  |
| 406 | P. O., Mentel, T. F., Wildt, J., Junninen, H., Jokinen, T., Kulmala, M., Worsnop, D. R., Thornton, J. A.,                                                                                                                                                                                                                                                                                                                                                                                                                                                                                                                                                                                                                                                                                                                                                                                                                                                                                                                                                                                                                                                                                                                                                                                                                                                                                                                                                                                                                                                                                                                                                                                                                                                                                                                                                                                                                                                                                                                                                                                                                        |
| 407 | Donahue, N., Kjaergaard, H. G., & Ehn, M. (2019). Highly Oxygenated Organic Molecules (HOM)                                                                                                                                                                                                                                                                                                                                                                                                                                                                                                                                                                                                                                                                                                                                                                                                                                                                                                                                                                                                                                                                                                                                                                                                                                                                                                                                                                                                                                                                                                                                                                                                                                                                                                                                                                                                                                                                                                                                                                                                                                      |
| 408 | from Gas-Phase Autoxidation Involving Peroxy Radicals: A Key Contributor to Atmospheric Aerosol.                                                                                                                                                                                                                                                                                                                                                                                                                                                                                                                                                                                                                                                                                                                                                                                                                                                                                                                                                                                                                                                                                                                                                                                                                                                                                                                                                                                                                                                                                                                                                                                                                                                                                                                                                                                                                                                                                                                                                                                                                                 |
| 409 | Chemical Reviews, 119(6), 3472–3509. https://doi.org/10.1021/acs.chemrev.8b00395                                                                                                                                                                                                                                                                                                                                                                                                                                                                                                                                                                                                                                                                                                                                                                                                                                                                                                                                                                                                                                                                                                                                                                                                                                                                                                                                                                                                                                                                                                                                                                                                                                                                                                                                                                                                                                                                                                                                                                                                                                                 |
| 410 | Birmili, W., Daniels, A., Bethke, R., Schechner, N., Brasse, G., Conrad, A., Kolossa-Gehring, M., Debiak, M.,                                                                                                                                                                                                                                                                                                                                                                                                                                                                                                                                                                                                                                                                                                                                                                                                                                                                                                                                                                                                                                                                                                                                                                                                                                                                                                                                                                                                                                                                                                                                                                                                                                                                                                                                                                                                                                                                                                                                                                                                                    |
| 411 | $Hurra \&, J., Uhde, E., Omelan, A., \& Salthammer, T. (2022). Formaldehyde, aliphatic aldehydes (C_2-1) aliphatic $ |
| 412 | $C_{11}$ ), furfural, and benzaldehyde in the residential indoor air of children and adolescents during the                                                                                                                                                                                                                                                                                                                                                                                                                                                                                                                                                                                                                                                                                                                                                                                                                                                                                                                                                                                                                                                                                                                                                                                                                                                                                                                                                                                                                                                                                                                                                                                                                                                                                                                                                                                                                                                                                                                                                                                                                      |
| 413 | German Environmental Survey 2014–2017 (GerES V). Indoor Air, 32(1).                                                                                                                                                                                                                                                                                                                                                                                                                                                                                                                                                                                                                                                                                                                                                                                                                                                                                                                                                                                                                                                                                                                                                                                                                                                                                                                                                                                                                                                                                                                                                                                                                                                                                                                                                                                                                                                                                                                                                                                                                                                              |
| 414 | https://doi.org/10.1111/ina.12927                                                                                                                                                                                                                                                                                                                                                                                                                                                                                                                                                                                                                                                                                                                                                                                                                                                                                                                                                                                                                                                                                                                                                                                                                                                                                                                                                                                                                                                                                                                                                                                                                                                                                                                                                                                                                                                                                                                                                                                                                                                                                                |
|     |                                                                                                                                                                                                                                                                                                                                                                                                                                                                                                                                                                                                                                                                                                                                                                                                                                                                                                                                                                                                                                                                                                                                                                                                                                                                                                                                                                                                                                                                                                                                                                                                                                                                                                                                                                                                                                                                                                                                                                                                                                                                                                                                  |

| 415 | $Brean, J., Beddows, D. \ C. \ S., Shi, Z., Temime-Roussel, B., Marchand, N., Querol, X., Alastuey, A., Minguill\'on, A., Minguillon, A., Minguillo$ |
|-----|----------------------------------------------------------------------------------------------------------------------------------------------------------------------------------------------------------------------------------------------------------------------------------------------------------------------------------------------------------------------------------------------------------------------------------------------------------------------------------------------------------------------------------------------------------------------------------------------------------------------------------------------------------------------------------------------------------------------------------------------------------------------------------------------------------------------------------------------------------------------------------------------------------------------------------------------------------------------------------------------------------------------------------------------------------------------------------------------------------------------------------------------------------------------------------------------------------------------------------------------------------------------------------------------------------------------------------------------------------------------------------------------------------------------------------------------------------------------------------------------------------------------------------------------------------------------------------------------------------------------------------------------------------------------------------------------------------------------------------------------------------------------------------------------------------------------------------------------------------------------------------------------------------------------------------------------------------------------------------------------------------------------------------------------------------------------------------------------------------------------------------------------------------------------------|
| 416 | M. C., & Harrison, R. M. (2020). Molecular insights into new particle formation in Barcelona, Spain.                                                                                                                                                                                                                                                                                                                                                                                                                                                                                                                                                                                                                                                                                                                                                                                                                                                                                                                                                                                                                                                                                                                                                                                                                                                                                                                                                                                                                                                                                                                                                                                                                                                                                                                                                                                                                                                                                                                                                                                                                                                                       |
| 417 | Atmospheric Chemistry and Physics, 20(16), 10029-10045. https://doi.org/10.5194/acp-20-10029-2020                                                                                                                                                                                                                                                                                                                                                                                                                                                                                                                                                                                                                                                                                                                                                                                                                                                                                                                                                                                                                                                                                                                                                                                                                                                                                                                                                                                                                                                                                                                                                                                                                                                                                                                                                                                                                                                                                                                                                                                                                                                                          |
| 418 | Brean, J., Harrison, R. M., Shi, Z., Beddows, D. C. S., Acton, W. J. F., Hewitt, C. N., Squires, F. A., & Lee, J.                                                                                                                                                                                                                                                                                                                                                                                                                                                                                                                                                                                                                                                                                                                                                                                                                                                                                                                                                                                                                                                                                                                                                                                                                                                                                                                                                                                                                                                                                                                                                                                                                                                                                                                                                                                                                                                                                                                                                                                                                                                          |
| 419 | (2019). Observations of highly oxidized molecules and particle nucleation in the atmosphere of                                                                                                                                                                                                                                                                                                                                                                                                                                                                                                                                                                                                                                                                                                                                                                                                                                                                                                                                                                                                                                                                                                                                                                                                                                                                                                                                                                                                                                                                                                                                                                                                                                                                                                                                                                                                                                                                                                                                                                                                                                                                             |
| 420 | Beijing. Atmospheric Chemistry and Physics, 19(23), 14933–14947. https://doi.org/10.5194/acp-19-                                                                                                                                                                                                                                                                                                                                                                                                                                                                                                                                                                                                                                                                                                                                                                                                                                                                                                                                                                                                                                                                                                                                                                                                                                                                                                                                                                                                                                                                                                                                                                                                                                                                                                                                                                                                                                                                                                                                                                                                                                                                           |
| 421 | 14933-2019                                                                                                                                                                                                                                                                                                                                                                                                                                                                                                                                                                                                                                                                                                                                                                                                                                                                                                                                                                                                                                                                                                                                                                                                                                                                                                                                                                                                                                                                                                                                                                                                                                                                                                                                                                                                                                                                                                                                                                                                                                                                                                                                                                 |
| 422 | Calogirou, A., Larsen, B. R., & Kotzias, D. (1999). Gas-phase terpene oxidation products: A review.                                                                                                                                                                                                                                                                                                                                                                                                                                                                                                                                                                                                                                                                                                                                                                                                                                                                                                                                                                                                                                                                                                                                                                                                                                                                                                                                                                                                                                                                                                                                                                                                                                                                                                                                                                                                                                                                                                                                                                                                                                                                        |
| 423 | Atmospheric Environment, 33(9), 1423-1439. https://doi.org/10.1016/S1352-2310(98)00277-5                                                                                                                                                                                                                                                                                                                                                                                                                                                                                                                                                                                                                                                                                                                                                                                                                                                                                                                                                                                                                                                                                                                                                                                                                                                                                                                                                                                                                                                                                                                                                                                                                                                                                                                                                                                                                                                                                                                                                                                                                                                                                   |
| 424 | Calvert, J. G., Mellouki, A., Orlando, J. J., Pilling, M. J., & Wallington, T. J. (2020). The Mechanisms of                                                                                                                                                                                                                                                                                                                                                                                                                                                                                                                                                                                                                                                                                                                                                                                                                                                                                                                                                                                                                                                                                                                                                                                                                                                                                                                                                                                                                                                                                                                                                                                                                                                                                                                                                                                                                                                                                                                                                                                                                                                                |
| 425 | Atmospheric Oxidation of the Oxygenates. Oxford University Press.                                                                                                                                                                                                                                                                                                                                                                                                                                                                                                                                                                                                                                                                                                                                                                                                                                                                                                                                                                                                                                                                                                                                                                                                                                                                                                                                                                                                                                                                                                                                                                                                                                                                                                                                                                                                                                                                                                                                                                                                                                                                                                          |
| 426 | https://doi.org/10.1093/oso/9780199767076.001.0001                                                                                                                                                                                                                                                                                                                                                                                                                                                                                                                                                                                                                                                                                                                                                                                                                                                                                                                                                                                                                                                                                                                                                                                                                                                                                                                                                                                                                                                                                                                                                                                                                                                                                                                                                                                                                                                                                                                                                                                                                                                                                                                         |
| 427 | Carlier, P., Hannachi, H., & Mouvier, G. (1986). The chemistry of carbonyl compounds in the atmosphere—A                                                                                                                                                                                                                                                                                                                                                                                                                                                                                                                                                                                                                                                                                                                                                                                                                                                                                                                                                                                                                                                                                                                                                                                                                                                                                                                                                                                                                                                                                                                                                                                                                                                                                                                                                                                                                                                                                                                                                                                                                                                                   |
| 428 | review. Atmospheric Environment (1967), 20(11), 2079–2099. https://doi.org/10.1016/0004-                                                                                                                                                                                                                                                                                                                                                                                                                                                                                                                                                                                                                                                                                                                                                                                                                                                                                                                                                                                                                                                                                                                                                                                                                                                                                                                                                                                                                                                                                                                                                                                                                                                                                                                                                                                                                                                                                                                                                                                                                                                                                   |
| 429 | 6981(86)90304-5                                                                                                                                                                                                                                                                                                                                                                                                                                                                                                                                                                                                                                                                                                                                                                                                                                                                                                                                                                                                                                                                                                                                                                                                                                                                                                                                                                                                                                                                                                                                                                                                                                                                                                                                                                                                                                                                                                                                                                                                                                                                                                                                                            |
| 430 | Cassanelli, P., Johnson, D., & Anthony Cox, R. (2005). A temperature-dependent relative-rate study of the OH                                                                                                                                                                                                                                                                                                                                                                                                                                                                                                                                                                                                                                                                                                                                                                                                                                                                                                                                                                                                                                                                                                                                                                                                                                                                                                                                                                                                                                                                                                                                                                                                                                                                                                                                                                                                                                                                                                                                                                                                                                                               |
| 431 | initiated oxidation of n-butane: The kinetics of the reactions of the 1- and 2-butoxy radicals. Physical                                                                                                                                                                                                                                                                                                                                                                                                                                                                                                                                                                                                                                                                                                                                                                                                                                                                                                                                                                                                                                                                                                                                                                                                                                                                                                                                                                                                                                                                                                                                                                                                                                                                                                                                                                                                                                                                                                                                                                                                                                                                   |
| 432 | Chemistry Chemical Physics, 7(21), 3702. https://doi.org/10.1039/b507137b                                                                                                                                                                                                                                                                                                                                                                                                                                                                                                                                                                                                                                                                                                                                                                                                                                                                                                                                                                                                                                                                                                                                                                                                                                                                                                                                                                                                                                                                                                                                                                                                                                                                                                                                                                                                                                                                                                                                                                                                                                                                                                  |
| 433 | Castañeda, R., Iuga, C., Álvarez-Idaboy, J. R., & Vivier-Bunge, A. (2012). Rate Constants and Branching                                                                                                                                                                                                                                                                                                                                                                                                                                                                                                                                                                                                                                                                                                                                                                                                                                                                                                                                                                                                                                                                                                                                                                                                                                                                                                                                                                                                                                                                                                                                                                                                                                                                                                                                                                                                                                                                                                                                                                                                                                                                    |
| 434 | Ratios in the Oxidation of Aliphatic Aldehydes by OH Radicals under Atmospheric Conditions. 56(3),                                                                                                                                                                                                                                                                                                                                                                                                                                                                                                                                                                                                                                                                                                                                                                                                                                                                                                                                                                                                                                                                                                                                                                                                                                                                                                                                                                                                                                                                                                                                                                                                                                                                                                                                                                                                                                                                                                                                                                                                                                                                         |
| 435 | 316–324.                                                                                                                                                                                                                                                                                                                                                                                                                                                                                                                                                                                                                                                                                                                                                                                                                                                                                                                                                                                                                                                                                                                                                                                                                                                                                                                                                                                                                                                                                                                                                                                                                                                                                                                                                                                                                                                                                                                                                                                                                                                                                                                                                                   |
| 436 | Chacon-Madrid, H. J., Presto, A. A., & Donahue, N. M. (2010). Functionalization vs. fragmentation: N-                                                                                                                                                                                                                                                                                                                                                                                                                                                                                                                                                                                                                                                                                                                                                                                                                                                                                                                                                                                                                                                                                                                                                                                                                                                                                                                                                                                                                                                                                                                                                                                                                                                                                                                                                                                                                                                                                                                                                                                                                                                                      |
| 437 | aldehyde oxidation mechanisms and secondary organic aerosol formation. Physical Chemistry                                                                                                                                                                                                                                                                                                                                                                                                                                                                                                                                                                                                                                                                                                                                                                                                                                                                                                                                                                                                                                                                                                                                                                                                                                                                                                                                                                                                                                                                                                                                                                                                                                                                                                                                                                                                                                                                                                                                                                                                                                                                                  |
| 438 | Chemical Physics, 12(42), 13975. https://doi.org/10.1039/c0cp00200c                                                                                                                                                                                                                                                                                                                                                                                                                                                                                                                                                                                                                                                                                                                                                                                                                                                                                                                                                                                                                                                                                                                                                                                                                                                                                                                                                                                                                                                                                                                                                                                                                                                                                                                                                                                                                                                                                                                                                                                                                                                                                                        |
| 439 | Ciccioli, P., Brancaleoni, E., Frattoni, M., Cecinato, A., & Brachetti, A. (1993). Ubiquitous occurrence of semi-                                                                                                                                                                                                                                                                                                                                                                                                                                                                                                                                                                                                                                                                                                                                                                                                                                                                                                                                                                                                                                                                                                                                                                                                                                                                                                                                                                                                                                                                                                                                                                                                                                                                                                                                                                                                                                                                                                                                                                                                                                                          |
| 440 | volatile carbonyl compounds in tropospheric samples and their possible sources. Atmospheric                                                                                                                                                                                                                                                                                                                                                                                                                                                                                                                                                                                                                                                                                                                                                                                                                                                                                                                                                                                                                                                                                                                                                                                                                                                                                                                                                                                                                                                                                                                                                                                                                                                                                                                                                                                                                                                                                                                                                                                                                                                                                |
| 441 | Environment. Part A. General Topics, 27(12), 1891–1901. https://doi.org/10.1016/0960-                                                                                                                                                                                                                                                                                                                                                                                                                                                                                                                                                                                                                                                                                                                                                                                                                                                                                                                                                                                                                                                                                                                                                                                                                                                                                                                                                                                                                                                                                                                                                                                                                                                                                                                                                                                                                                                                                                                                                                                                                                                                                      |
| 442 | 1686(93)90294-9                                                                                                                                                                                                                                                                                                                                                                                                                                                                                                                                                                                                                                                                                                                                                                                                                                                                                                                                                                                                                                                                                                                                                                                                                                                                                                                                                                                                                                                                                                                                                                                                                                                                                                                                                                                                                                                                                                                                                                                                                                                                                                                                                            |

| 443 | Crounse, J. D., Nielsen, L. B., Jørgensen, S., Kjaergaard, H. G., & Wennberg, P. O. (2013). Autoxidation of                                                                                                                                                                                                                                                                                                                                                                                                                                                                                                                                                                                                                                                                                                                                                                                                                                                                                                                                                                                                                                                                                                                                                                                                                                                                                                                                                                                                                                                                                                                   |
|-----|-------------------------------------------------------------------------------------------------------------------------------------------------------------------------------------------------------------------------------------------------------------------------------------------------------------------------------------------------------------------------------------------------------------------------------------------------------------------------------------------------------------------------------------------------------------------------------------------------------------------------------------------------------------------------------------------------------------------------------------------------------------------------------------------------------------------------------------------------------------------------------------------------------------------------------------------------------------------------------------------------------------------------------------------------------------------------------------------------------------------------------------------------------------------------------------------------------------------------------------------------------------------------------------------------------------------------------------------------------------------------------------------------------------------------------------------------------------------------------------------------------------------------------------------------------------------------------------------------------------------------------|
| 444 | Organic Compounds in the Atmosphere. The Journal of Physical Chemistry Letters, 4(20), 3513–3520.                                                                                                                                                                                                                                                                                                                                                                                                                                                                                                                                                                                                                                                                                                                                                                                                                                                                                                                                                                                                                                                                                                                                                                                                                                                                                                                                                                                                                                                                                                                             |
| 445 | https://doi.org/10.1021/jz4019207                                                                                                                                                                                                                                                                                                                                                                                                                                                                                                                                                                                                                                                                                                                                                                                                                                                                                                                                                                                                                                                                                                                                                                                                                                                                                                                                                                                                                                                                                                                                                                                             |
| 446 | Duan, J., Tan, J., Yang, L., Wu, S., & Hao, J. (2008). Concentration, sources and ozone formation potential of                                                                                                                                                                                                                                                                                                                                                                                                                                                                                                                                                                                                                                                                                                                                                                                                                                                                                                                                                                                                                                                                                                                                                                                                                                                                                                                                                                                                                                                                                                                |
| 447 | volatile organic compounds (VOCs) during ozone episode in Beijing. Atmospheric Research, 88(1),                                                                                                                                                                                                                                                                                                                                                                                                                                                                                                                                                                                                                                                                                                                                                                                                                                                                                                                                                                                                                                                                                                                                                                                                                                                                                                                                                                                                                                                                                                                               |
| 448 | 25-35. https://doi.org/10.1016/j.atmosres.2007.09.004                                                                                                                                                                                                                                                                                                                                                                                                                                                                                                                                                                                                                                                                                                                                                                                                                                                                                                                                                                                                                                                                                                                                                                                                                                                                                                                                                                                                                                                                                                                                                                         |
| 449 | Ehn, M., Thornton, J. A., Kleist, E., Sipilä, M., Junninen, H., Pullinen, I., Springer, M., Rubach, F., Tillmann,                                                                                                                                                                                                                                                                                                                                                                                                                                                                                                                                                                                                                                                                                                                                                                                                                                                                                                                                                                                                                                                                                                                                                                                                                                                                                                                                                                                                                                                                                                             |
| 450 | R., Lee, B., Lopez-Hilfiker, F., Andres, S., Acir, IH., Rissanen, M., Jokinen, T., Schobesberger, S.,                                                                                                                                                                                                                                                                                                                                                                                                                                                                                                                                                                                                                                                                                                                                                                                                                                                                                                                                                                                                                                                                                                                                                                                                                                                                                                                                                                                                                                                                                                                         |
| 451 | Kangasluoma, J., Kontkanen, J., Nieminen, T., Mentel, T. F. (2014). A large source of low-volatility                                                                                                                                                                                                                                                                                                                                                                                                                                                                                                                                                                                                                                                                                                                                                                                                                                                                                                                                                                                                                                                                                                                                                                                                                                                                                                                                                                                                                                                                                                                          |
| 452 | secondary organic aerosol. Nature, 506(7489), 476–479. https://doi.org/10.1038/nature13032                                                                                                                                                                                                                                                                                                                                                                                                                                                                                                                                                                                                                                                                                                                                                                                                                                                                                                                                                                                                                                                                                                                                                                                                                                                                                                                                                                                                                                                                                                                                    |
| 453 | Fan, C., Yan, H., Wang, W., Sun, Z., & Ge, M. (2024). Study on the reactions of n-pentanal and n-hexanal with                                                                                                                                                                                                                                                                                                                                                                                                                                                                                                                                                                                                                                                                                                                                                                                                                                                                                                                                                                                                                                                                                                                                                                                                                                                                                                                                                                                                                                                                                                                 |
| 454 | Br atoms: Kinetics, gas-phase products, and SOA formation. Atmospheric Environment, 339, 120869.                                                                                                                                                                                                                                                                                                                                                                                                                                                                                                                                                                                                                                                                                                                                                                                                                                                                                                                                                                                                                                                                                                                                                                                                                                                                                                                                                                                                                                                                                                                              |
| 455 | https://doi.org/10.1016/j.atmosenv.2024.120869                                                                                                                                                                                                                                                                                                                                                                                                                                                                                                                                                                                                                                                                                                                                                                                                                                                                                                                                                                                                                                                                                                                                                                                                                                                                                                                                                                                                                                                                                                                                                                                |
| 456 | Fry, J. L., Draper, D. C., Barsanti, K. C., Smith, J. N., Ortega, J., Winkler, P. M., Lawler, M. J., Brown, S. S.,                                                                                                                                                                                                                                                                                                                                                                                                                                                                                                                                                                                                                                                                                                                                                                                                                                                                                                                                                                                                                                                                                                                                                                                                                                                                                                                                                                                                                                                                                                            |
| 457 | Edwards, P. M., Cohen, R. C., & Lee, L. (2014). Secondary Organic Aerosol Formation and Organic                                                                                                                                                                                                                                                                                                                                                                                                                                                                                                                                                                                                                                                                                                                                                                                                                                                                                                                                                                                                                                                                                                                                                                                                                                                                                                                                                                                                                                                                                                                               |
| 458 | Nitrate Yield from NO <sub>3</sub> Oxidation of Biogenic Hydrocarbons. Environmental Science & Technology,                                                                                                                                                                                                                                                                                                                                                                                                                                                                                                                                                                                                                                                                                                                                                                                                                                                                                                                                                                                                                                                                                                                                                                                                                                                                                                                                                                                                                                                                                                                    |
| 459 | 48(20), 11944–11953. https://doi.org/10.1021/es502204x                                                                                                                                                                                                                                                                                                                                                                                                                                                                                                                                                                                                                                                                                                                                                                                                                                                                                                                                                                                                                                                                                                                                                                                                                                                                                                                                                                                                                                                                                                                                                                        |
| 460 | Hallquist,M.,Wenger,J.C.,Baltensperger,U.,Rudich,Y.,Simpson,D.,Claeys,M.,Dommen,J.,Donahue,N.Rudich,Y.,Simpson,D.,Claeys,M.,Dommen,J.,Donahue,N.Rudich,Y.,Simpson,D.,Claeys,M.,Dommen,J.,Donahue,N.Rudich,Y.,Simpson,D.,Claeys,M.,Dommen,J.,Donahue,N.Rudich,Y.,Simpson,D.,Claeys,M.,Dommen,J.,Donahue,N.Rudich,Y.,Simpson,D.,Claeys,M.,Dommen,D.,Donahue,N.Rudich,P.Rudich,P.Rudich,P.Rudich,P.Rudich,P.Rudich,P.Rudich,P.Rudich,P.Rudich,P.Rudich,P.Rudich,P.Rudich,P.Rudich,P.Rudich,P.Rudich,P.Rudich,P.Rudich,P.Rudich,P.Rudich,P.Rudich,P.Rudich,P.Rudich,P.Rudich,P.Rudich,P.Rudich,P.Rudich,P.Rudich,P.Rudich,P.Rudich,P.Rudich,P.Rudich,P.Rudich,P.Rudich,P.Rudich,P.Rudich,P.Rudich,P.Rudich,P.Rudich,P.Rudich,P.Rudich,P.Rudich,P.Rudich,P.Rudich,P.Rudich,P.Rudich,P.Rudich,P.Rudich,P.Rudich,P.Rudich,P.Rudich,P.Rudich,P.Rudich,P.Rudich,P.Rudich,P.Rudich,P.Rudich,P.Rudich,P.Rudich,P.Rudich,P.Rudich,P.Rudich,P.Rudich,P.Rudich,P.Rudich,P.Rudich,P.Rudich,P.Rudich,P.Rudich,P.Rudich,P.Rudich,P.Rudich,P.Rudich,P.Rudich,P.Rudich,P.Rudich,P.Rudich,P.Rudich,P.Rudich,P.Rudich,P.Rudich,P.Rudich,P.Rudich,P.Rudich,P.Rudich,P.Rudich,P.Rudich,P.Rudich,P.Rudich,P.Rudich,P.Rudich,P.Rudich,P.Rudich,P.Rudich,P.Rudich,P.Rudich,P.Rudich,P.Rudich,P.Rudich,P.Rudich,P.Rudich,P.Rudich,P.Rudich,P.Rudich,P.Rudich,P.Rudich,P.Rudich,P.Rudich,P.Rudich,P.Rudich,P.Rudich,P.Rudich,P.Rudich,P.Rudich,P.Rudich,P.Rudich,P.Rudich,P.Rudich,P.Rudich,P.Rudich,P.Rudich,P.Rudich,P.Rudich,P.Rudich,P.Rudich,P.Rudich,P.Rudich,P.Rudich,P.Rudich,P.Rudich,P.Rudich,P.Rudich,P.Rudich,P.Rudich,P.Rudich,P.Rudich,P.Ru |
| 461 | M., George, C., Goldstein, A. H., Hamilton, J. F., Herrmann, H., Hoffmann, T., Iinuma, Y., Jang, M.,                                                                                                                                                                                                                                                                                                                                                                                                                                                                                                                                                                                                                                                                                                                                                                                                                                                                                                                                                                                                                                                                                                                                                                                                                                                                                                                                                                                                                                                                                                                          |
| 462 | Jenkin, M. E., Jimenez, J. L., Kiendler-Scharr, A., Maenhaut, W., Wildt, J. (2009). The formation,