# Peer review of "Rapid formation of secondary aerosol precursors from the autoxidation of C5-C8 n-1 aldehydes 2 Shawon Barua1,\*, Avinash Kumar1, Prasenjit Seal1, Siddharth Iyer1, and Matti Rissanen1,2,\* 3 1Aerosol Physics Laboratory, Physics Unit, F"

_EGUsphere, 2025_

## Referee Comment (RC5)

The manuscript Barua et. al. reports the formation of HOMs due to oxidation of C5-C8 aldehydes, with their flow tube setup and subsequent analysis via mass spectrometry. Length of the aldehydic carbon chain directly affected the oxidation products. Molecules with longer carbon atoms produced HOMs with higher intensities. Addition of NO and its effect on HOM accretion products are discussed for different concentrations of NO.

General comments:

The experiments are well designed, results are clearly shown, and the manuscript is structured properly. The relevance of aldehydes and understanding their role in SOA formation is important for atmospheric chemistry research. The manuscript could be enhanced by adding some statistical insights perhaps (e.g. linear regression tests) on the observed trends, provided the dataset permits statistical analysis.

Besides that, here are some 'Specific comments' for consideration:

L29: '…to the aerosol material that is formed by….'

Firstly, the term 'particle' could be used instead of 'material'.

Secondly, from this sentence it seems oxidation of VOCs is the only method for SOA formation. In L39, the authors mention '…its sources and formation processes are yet to be fully understood.' It would be informative for the reader to know what other processes can lead to the formation of SOA. Not necessarily in detail but stating the other processes in the beginning might be useful for the reader.

L56-61: 'In high NOx (NO + NO2) condition…' Splitting this sentence would make the text easier to follow.

L62-65: 'Besides, it can also produce…' same as the previous comment. Splitting the sentence with their respective references is suggested.

L89: ' …competitive for autoxidation reaction chain propagation…' this phrase seems too long winded. Some alternative suggestions for rephrasing: "…competitive with chain propagation in autoxidation…" or "…competitive with autoxidation chain-propagation reactions…". Either usage depends on which aspect of the phrase the authors want to emphasize on.

L109: 'Fig. S1 in the Supplement…' It might be convenient for a reader who is not directly related to such experiments, to have the experimental schematic or a flowchart appear in the main manuscript, rather than in a supplement.

L139: omit 'also'

Some information on the data processing steps, e.g. baseline corrections, would be helpful for reproducibility. They could be added at the end of the Method section, where data analysis is mentioned.

L180: comma after H-shift.

L181-184: Splitting the sentence is suggested after 'while'.

L200: Typo 'withing'

L205-206: '....while heptanal produced most oxygenated products are C7H12–14O8 (see Figure 2C)'. Grammatically incorrect. Kindly rephrase.

L307: References should be in ascending chronological order. Also, in L42, L 64, L330, etc., and many other instances, the reference order is mixed up. Kindly check throughout.

L356: '...various NO conditions'. Could the conditions be specified here briefly, for which conditions the oxidation products were enhanced.